# Impact of Bevacizumab on Visual Function, Tumor Size, and Toxicity in Pediatric Progressive Optic Pathway Glioma: A Retrospective Nationwide Multicentre Study

**DOI:** 10.3390/cancers14246087

**Published:** 2022-12-10

**Authors:** Carlien A. M. Bennebroek, Judith van Zwol, Giorgio L. Porro, Rianne Oostenbrink, Anne T. M. Dittrich, Annabel L. W. Groot, Jan W. Pott, Etienne J. M. Janssen, Noël J. Bauer, Maria M. van Genderen, Peerooz Saeed, Maarten H. Lequin, Pim de Graaf, Antoinette Y. N. Schouten-van Meeteren

**Affiliations:** 1Department of Ophthalmology, Amsterdam UMC Location University of Amsterdam, 1053 VE Amsterdam, The Netherlands; 2Cancer Center Amsterdam, Cancer Treatment and Quality of Life, 1081 HV Amsterdam, The Netherlands; 3Department of Ophthalmology Utrecht UMC, 3584 CX Utrecht, The Netherlands; 4ENCORE-NF1 Center, Department of General Pediatrics, Erasmus MC, 3015 GD Rotterdam, The Netherlands; 5Department of Pediatrics, Radboud University Medical Center, Amalia Children’s Hospital, 6525 GA Nijmegen, The Netherlands; 6Department of Ophthalmology, Radboud University Medical Center, 6525 GA Nijmegen, The Netherlands; 7Department of Ophthalmology, University Medical Center Groningen, University of Groningen, 9713 GZ Groningen, The Netherlands; 8Department of Neurology, Maastricht University Medical Center, 6221 CZ Maastricht, The Netherlands; 9Department of Ophthalmology, Maastricht University Medical Center, 6229 HX Maastricht, The Netherlands; 10Diagnostic Center for Complex Visual Disorders, Bartiméus, 3703 AJ Zeist, The Netherlands; 11Department of Radiology, Princess Máxima Center for Pediatric Oncology, 3584 CS Utrecht, The Netherlands; 12Department of Radiology and Nuclear Medicine, Amsterdam UMC Location Vrije Universiteit Amsterdam, 1081 HV Amsterdam, The Netherlands; 13Cancer Center Amsterdam, Imaging and Biomarkers, 1081 HV Amsterdam, The Netherlands; 14Department of Neuro-Oncology, Princess Máxima Center for Pediatric Oncology, 3584 CS Utrecht, The Netherlands

**Keywords:** bevacizumab, optic pathway glioma, low grade glioma, child, systemic anticancer therapy, visual function, visual acuity, visual outcome, progression, toxicity

## Abstract

**Simple Summary:**

Around 30% of children with optic pathway glioma (OPG) require next order systemic anticancer therapy (SAT) in case of progression. Bevacizumab (BVZ) is considered an effective subsequent SAT for pediatric low grade glioma. This retrospective nationwide multicentre study evaluated the effect of treatment with BVZ of 33 children with OPG. A new finding is that visual acuity stabilised (74.4%) or improved (20.5%) after treatment in 39 analysed eyes, visual field stabilised in 15.4% and improved in 73.1% of 25 eyes. Progression free survival from start of treatment decreased from 70.9% at 18 months to 38.0% at 36 months. Reversible severe toxicity was observed in five of 33 patients (15.2%). Our results suggest that the majority of patients with OPG treated with BVZ temporarily stabilize, however most show progression at a later time point. Visual functions, especially visual field, improve in a high percentage of patients. BVZ treatment is suggested to be a useful successive SAT.

**Abstract:**

Backgrounds: Bevacizumab (BVZ) is used as a subsequent line of treatment for pediatric optic pathway glioma (OPG) in the case of progression. Data on the treatment effect concerning tumor progression and visual function are scarce and nationwide studies are lacking. Methods: We performed a retrospective, nationwide, multicentre cohort study including all pediatric patients with OPG treated with BVZ in the Netherlands (2009–2021). Progression-free survival, change in visual acuity and visual field, MRI-based radiologic response, and toxicity were evaluated. Results: In total, 33 pediatric patients with OPG were treated with BVZ (median 12 months). Visual acuity improved in 20.5%, remained stable in 74.4%, and decreased in 5.1% of 39 of all analysed eyes. The monocular visual field improved in 73.1%, remained stable in 15.4%, and decreased in 7.7% of 25 analysed eyes. Radiologic response at the end of therapy showed a partial response in 7 patients (21.9%), minor response in 7 (21.9%), stable disease in 15 (46.9%), and progressive disease in 3 (9.3%). Progression-free survival at 18 and 36 months after the start of BVZ reduced from 70.9% to 38.0%. Toxicity (≥grade 3 CTCAE) during treatment was observed in five patients (15.2%). Conclusion: Treatment of BVZ in pediatric patients with OPG revealed stabilisation in the majority of patients, but was followed by progression at a later time point in more than 60% of patients. This profile seems relatively acceptable given the benefits of visual field improvement in more than 70% of analysed eyes and visual acuity improvement in more than 20% of eyes at the cessation of BVZ.

## 1. Introduction

Pediatric optic pathway glioma (OPG) concerns a low-grade glioma (LGG) (WHO grade 1) confined to the optic pathway, histologically represented by pilocytic astrocytoma (PA) in 90% of cases. In the case of tumor growth and/or clinical deterioration, systemic anticancer treatment (SAT), mostly chemotherapy, is often preferred over radiotherapy to avoid endocrine, vascular, and/or cognitive damage [1].

In the Netherlands, initial systemic therapy is mostly represented by carboplatin and vincristine, followed by vinblastine in the case of progression [2]. Bevacizumab (BVZ) combined with irinotecan (IRI) was introduced in 2009 as the next line of treatment for progressive LGG [3]. As a humanized monoclonal antibody, BVZ is a potent vascular endothelial growth factor (VEGF) inhibitor due to its highly specific binding capacity to VEGF isoforms [4]. Histological studies on PA substantiate the antiangiogenic effect of BVZ by the presence of the vascular endothelial growth factor receptor 2 (VEGFR−2) on PA endothelium [5]. Thus far, the majority of studies on pediatric LGG at diverse anatomic locations have evaluated treatment outcome after the combination of BVZ and IRI [3,6,7,8,9,10], diverse combined strategies [11], or BVZ monotherapy [12,13]. Sample sizes vary from 7–88 LGG.

Treatment of LGG with BVZ tends to stabilize tumor volume, but progression frequently occurs after discontinuation of BVZ (14–93%) within a median of 5–12 months after cessation of BVZ [7,8,9,11,12,14]. Studies on the toxicity profile are reported and show an acceptable profile of grade 3 toxicity according to CTCAE criteria (Common Terminology Criteria for Adverse Events) [8,10,11,12], which is reversible after cessation [8,11]. However, Green et al. presented serious grade 4 toxicity in 5% [10].

Evaluation of the treatment effect on OPG requires twofold analyses containing evaluation of visual function and radiologic response (according to the RAPNO criteria (Response Assessment in Pediatric Neuro-Oncology)) [15]. Currently, no correlation among both outcome parameters is shown [16,17]. Bevacizumab has shown a promising effect on visual function, with improvement or stabilisation of visual acuity of 73% of eyes with treated OPGs [10].

In the past era, drugs targeting the molecular biologic changes in the MAP-kinase pathway became successful for LGG treatment [18,19]. The activation of this pathway is caused by activation of the BRAF-oncogene, in pediatric OPG frequently represented by the KIAA1549-BRAF fusion, which can be identified in 50–60% of OPGs [20,21]. The 5-year progression-free survival (PFS) of OPG with KIAA1549-BRAF fusion is superior to non-fusion OPG after first line SAT [20]. Thus far, knowledge of OPG progression after SAT with or without KIAA1549-BRAF fusion is lacking.

We evaluated the efficacy and safety of BVZ for OPG in a nationwide cohort of patients by analysing visual function’s best corrected visual acuity (BCVA) and visual field (VF), tumor size, toxicity profile, and progression-free survival.

## 2. Materials and Methods

### 2.1. Study Design and Data Collection

This is a retrospective nationwide multicentre study. All consecutive children (0–17 year) in the Netherlands diagnosed with an OPG (Modified Dodge Classification (MDC) stage 1–4) [22] and treated with BVZ during at least one month, with or without other chemotherapy, were eligible for this study. The diagnosis of OPG with or without neurofibromatosis type 1 (NF1) was confirmed by its appearance on MRI and/or tissue analysis obtained via surgery or biopsy.

The study was approved by the Dutch Childhood Oncology Group (DCOG). Ethical committees of participating centers (Princess Máxima Center (the national tertiary pediatric oncology center), Amsterdam UMC, Erasmus MC, University Medical Center Utrecht (UMCU), Radboud university medical center, UMC Groningen, Maastricht University Medical Center) and visual rehabilitation centers (Stichting Bartiméus and Visio) in the Netherlands gave approval for collection of coded data. Patients were retrieved via the registry of the national database of the DCOG (diagnosed with OPG from January 2003 until December 2018) and all patients from the Princess Máxima Center (diagnosed with OPG from January 2019 until December 2021). In addition, pediatric ophthalmologists and oncologists from all Dutch university hospitals were consulted to identify potentially non-registered patients since 2003. For 32 patients follow-up was continued until 18 June 2022, and for one patient until 31 December 2020.

Informed consent was given by patients and/or parents or legal guardians registered at the DCOG, and it was additionally required at two participating centers, namely UMCU and Princess Máxima Center. An opt-out procedure was offered to patients registered in local databases at the Amsterdam UMC. Other centers provided permission to use coded patient data by waiver of consent.

### 2.2. Clinical Data Collection

Data were collected by reviewing medical records on patient characteristics, including sex, NF1 status, age at diagnosis, age at start of initial treatment and start of BVZ, previous treatment, indication for start of BVZ (ophthalmological/radiological), number of treatment cycles, and dosage of BVZ and concurrent systemic anticancer therapy (SAT). If tumor biopsy had been performed, the histology and BRAF V600E mutation and KIAA1549-BRAF fusion were registered.

### 2.3. Response Evaluation

#### 2.3.1. Outcomes

The primary endpoint consisted of a combination of outcome parameters analysing the treatment effect of BVZ. These parameters were: change in best corrected visual acuity (BCVA), visual field (VF), radiologic response, and progression-free survival. A secondary endpoint was the toxicity of BVZ.

#### 2.3.2. Visual Function

Visual functions were extracted from medical records by collecting BCVA and VF performed within 2 months before start of BVZ and within 3 months after cessation of BVZ. Monocular BCVA was registered from age-appropriate testing methods (Teller Acuity Cards, Cardiff Acuity Test, Kays Pictures, Snellen charts) and converted into the logarithm of minimal angle of resolution scale (LogMAR) for statistical purposes. Binocular visual impairment was categorized according to the definitions of visual impairment and blindness as described by the World Health Organization: mild or no visual impairment (BCVA ≤ 0.5 LogMAR), moderate visual impairment (BCVA > 0.5 ≤ 1.0 LogMAR), severe visual impairment (BCVA >1.0 ≤ 1.3 LogMAR), and blindness (BCVA > 1.3 LogMAR) [23]. Change in BCVA was defined as a change of ≥ or ≤0.2 LogMAR. Visual acuity values corresponding to counting fingers, hand motion, light perception, and no light perception were converted to 2.0, 2.4, 2.7, and 3.0 LogMAR, respectively [24].

Available age-adapted VF testing methods included the Behavioral Visual Field (BEFIE) Screening test [25], the semiautomatic-static Peritest [26], Goldmann kinetic perimetry, or the automatic Humphrey Visual Field Analyzer (HFA). Visual fields were blinded and evaluated by two independent experienced (pediatric (CB) and neuro- (GP)) ophthalmologists for abnormalities and change. Discrepancies between graders were resolved by discussion and mutual agreement. Cases of HFA low reliability, defined by test-specific cut-off values (i.e., HFA 30–2: false-positive errors, false-negative errors or fixation losses greater or equal to 20%), were excluded from further analyses. The following items were scored: scotoma (central/paracentral/cecocentral), quadrant or hemianopia (partial/absolute), and location of defects (nasal/temporal/central). The change in VF defects was scored as: any change in visual field according to the clinical judgement of both assessors (BEFIE and Goldmann kinetic perimetry), change of ≥ 3 consecutive significant defects (*p* < 0.05) (HFA and Peritest).

#### 2.3.3. Radiologic Response

Radiologic analyses were independently performed by two experienced neuroradiologists (ML and PG). The anatomic location of OPG was classified following the MDC [22] (stage 1: optic nerve(s), stage 2: chiasm, stage 3: optic tract, stage 4: posterior optic tract, presence of hypothalamic involvement and leptomeningeal dissemination).

Radiologic tumor size and response evaluation, according to the RAPNO criteria [15], was performed on the MRI obtained prior to the start of BVZ and the most recent MRI after cessation of BVZ. The RAPNO working group has defined recommendations for radiologic response criteria specifically designed for pediatric LGG [15]. Response assessment was performed by calculation of the product of 3 perpendicular measurements on T1 with contrast enhancement and/or T2-FLAIR. Both cystic and solid compartments were included in the measurements, according to the recommendations in the published guideline [15]. Diffuse OPG were divided into two subcategories: involvement versus non-involvement of the optic tract. The response categories applied in this study are defined as: complete response: complete disappearance of the OPG; partial response: ≥ 50% decrease; minor response: 25–49% decrease; stable disease: 24% decrease to 25% increase; progressive disease: >25% increase of OPG [15].

#### 2.3.4. Toxicity

The toxicity profile of BVZ, from grade 2 onwards, was scored based on documentation in patients records from start of treatment until three months after cessation, according to the pediatric-specific criteria of the Common Terminology Criteria for Adverse Events (CTCAE v5.0) [27]. As a majority of patients received a combination of SAT (in addition to BVZ), all results of blood samples obtained in the period from start to cessation were scored on abnormalities in blood count, liver, and renal function.

#### 2.3.5. Progression

Progression, as defined by the local medical team, was scored as progression due to increase of tumor size, decrease of BCVA and/or VF, or new dissemination. In cases of progression, successive therapy was scored. Time to progression was measured by comparison between start of BVZ to the earliest date of progression. Progression-free survival was determined at 18 and 36 months after start of BVZ, which can be translated to 6 and 24 months after the intended completion of 12 months of BVZ. Patients were considered censored in case of death or having had a follow-up of less than 36 months.

Comparison of PFS was made among the NF1 and KIAA1549-BRAF fusion population, the four age categories (0–2 yr/3–5 yr/6–9 yr/10–18 yr), and diverse treatment combinations.

### 2.4. Data Analysis

Data were collected by creating electronic case report forms in Castor (https://www.castoredc.com/(accessed on 3 November 2022)) and exporting them to SPSS software for Windows (version 26.0.0.1, SPSS Inc., Chicago, IL, USA) for statistical analyses. Data analysis was performed using descriptive statistics. Continuous variables were presented by mean and standard deviation (in case of normal distribution) or median range and interquartile range and categorical data by frequency and percentage. For continuous variables, differences between groups were tested with the Student’s t test for normally distributed data or Mann–Whitney U test for non-normally distributed data.

The 18- and 36-month PFS with 95% confidence interval (CI) after the start of BVZ were calculated with the Kaplan–Meier method for the total population. Stratified comparison of the PFS was performed by applying log rank analysis.

## 3. Results

This nationwide study cohort consisted of 33 patients, who started BVZ between December 2009 and December 2021. Initially, 35 patients with OPG treated with BVZ were identified. One patient was excluded due to lacking data on treatment schedule, toxicity of BVZ, and visual function, and another single patient received one dosage of BVZ and immediately stopped treatment because of suspected severe allergy.

### 3.1. Baseline Characteristics

The study cohort contained 20 males (60.6%) and 13 females (39.4%). Thirteen patients had NF1 (39.4%). The baseline characteristics are presented in Table 1. Five OPGs were located solely in the chiasm. A total of 28 OPGs (84.8%) presented with diffuse spreading along the optic pathway along diverse stages (stage 1–4 MDC) (Appendix A), of which 24 involved spreading in the posterior pathway (≥MDC stage 3). In 17 (51.5%) patients (3 with NF1, 14 without NF1) a biopsy was performed, of which 13 (76.5%) revealed a pilocytic astrocytoma (PA). Twenty-eight OPGs (84.8%) showed hypothalamic involvement, and in six patients (18%) leptomeningeal metastases were present. BRAF V600E evaluation was available in 12 biopsies and revealed no mutations. KIAA1549-BRAF fusion was present in 9 of 15 samples analysed for this fusion: 8 samples were a PA and 1 a pilomyxoid astrocytoma.

Patients were diagnosed with OPG at a median age of 2.4 years (range: 0.3–10.2 years) and started BVZ therapy at a median age of 7.2 years (range: 0.3–10.2 years). The median number of prior episodes of SAT was two (range: 0–4 episodes). The diverse types of (prior) therapy applied per treatment phase are presented in Appendix A. Treatment was initiated due to radiologic progression in 19 patients (57.6%), decrease of visual function in 12 (36.4%), a combined deterioration in 1 patient (3.0%), and presence of metastases in 1 patient (3.0%). A total of 33 patients received a median of 26 (range 4–89) doses of BVZ (10 mg/kg, started every 2 weeks), combined with IRI in 27 patients (81.8%) (IRI: median of 16 doses (range 4–54)) (125 mg/m2, started every 2 weeks) and combined with vinblastine (VBL) in 5 patients (15.2%) (VBL: median 3 mg/m2, weekly) (median of 32 doses (range 9–66)). All individual treatment schedules are presented in Appendix A. As treatment cycles show a variability in the combination of BVZ with or without IRI or VBL, further communication in this article is represented by ‘BVZ’. A total of 10 patients had prolonged BVZ treatment after 26 doses in 12 months (median doses: 36.5 (range 30–89), which is considered the current standard, according to the Dutch recommendations for treatment of LGG with BVZ. The median follow-up after start of BVZ was 40 months (range 6–150 months).

A single patient was still on BVZ treatment at the end of follow-up (total of 13 doses BVZ (Appendix A). Treatment was cessated prematurely in 8 patients: in 5 due to radiologic progression of OPG (median 6 months after start (range 2–8 months) and in 3 due to side effects.

### 3.2. Visual Function

At the start of therapy, 3 children (11.5%) had binocular moderate visual impairment (BCVA > 0.5–≤ 1.0 LogMAR), 2 (7.7%) had severe impairment (BCVA >1.0–≤ 1.3 LogMAR), and 7 (26.9%) were considered blind (BCVA > 1.3 LogMAR), according to the WHO criteria of visual impairment and blindness [23]). Three children were blind in both eyes (LogMAR 3.0) at start of treatment, while seven other patients were blind in one eye (see Table 2), and BCVA did not improve with time in any blind eye at follow up. Monocular data of BCVA and VF were absent in 7 patients (21.2%), with a median age of 1.2 year (range 1.1—3.3 years). A cause for absence of registration was a frequently found item: limited cooperation at young age.

Monocular evaluation of the treatment effect of BVZ on BCVA (≥ or ≤0.2 LogMAR) was performed in 39 eyes of 23 patients (eyes with LogMAR 3.0 excluded). The median BCVA at start was 0.2 LogMAR (range −0.10 to 1.8 LogMAR). While 8 of the 39 eyes (20.5%) showed improvement of BCVA after the end of treatment, 29 eyes (74.4%) remained stable and 2 eyes (5.1%) deteriorated. After cessation, the median BCVA was 0.32 LogMAR (range: −0.08–3.0 LogMAR).

Visual fields at the start and after cessation of BVZ were obtained from 15 children: VF of 1 eye in 5 patients and of both eyes in 10 patients. The BEFIE test was performed in 2 patients (13.3%), Peritest in 8 (53.3%), Goldmann in 2 (13.3%) and HFA in 3 (20.0%). The median age at start of therapy of patients for whom a visual field was available was 9.9 years (range 2.3–17.7 years), compared to a median 3.8 years (range 1.1–12.0 years) for patients with no available VF (exclusion of patients with BCVA of LogMAR ≥ 2.7). No VF met the criteria of visual impairment (constriction ≤ 30 degrees) or blindness (constriction ≤ 10 degrees) according to the WHO.

Monocular change of VF was evaluated in 26 eyes. A total of 19 of 26 eyes (73.1%) showed improvement, 4 eyes (15.4%) remained stable, 2 eyes (7.7%) showed deterioration, and in 1 eye (3.8%) the VF defect shifted from one quadrant to another.

Baseline data and effects in change of BCVA and VF are presented in Table 2. Simultaneous evaluation of BCVA and VF could be performed in 20 eyes of 14 patients. Combined improvement of BCVA and VF appeared in 5 eyes, stable BCVA and improved VF in 11 eyes, BCVA decrease and VF improvement in 1 eye, both stable BCVA and VF in 1 eye, and stable BCVA and decrease VF in 2 eyes.

All individual data on BCVA and VF are reported for each case separately in Appendix A.

### 3.3. Radiologic Evaluation

MRI data for radiologic response assessment (RAPNO) within 3 months after cessation of treatment were available in 32 patients (97.0%). A total of 7 patients with OPG (21.9%) showed a partial response, 7 (21.9%) showed a minor response, 15 (46.9%) OPG remained stable, and 3 OPG showed progressive disease (9.3%). In more than 50% of diffuse OPG (involvement > 1 MDC stage), manual measurements of tumor dimension in 3 perpendicular lines could not be performed due to the irregular pattern of OPG in the individual MDC sub location. No separate measurements were performed on the diverse MDC sub locations, but 3 perpendicular measurements at the level of the chiasm were performed instead.

### 3.4. Toxicity Profile

Toxicity grade 2 or 3 (CTCAE) occurred in 12 of 33 patients (36.4%) during BVZ therapy: grade 2 toxicity occurred in 7 patients (21.2%) and ≥grade 3 in 5 patients (15.2%) (see Table 3). In 5 patients (15.5%), 2 different side effects of ≥grade 2 were observed simultaneously. Side effects (≥grade 2) appeared after median 6.9 months after start of treatment (range 0–40 months) (median 15 doses (range 2–84)), which required cessation of BVZ in 83.4%. Grade 3 toxicity appeared after median 9 weeks (range 4–24 weeks) of treatment. A total of 3 patients stopped treatment prematurely after median 6.5 doses (range 4–9 doses) of BVZ due to side effects (colitis grade 2 CTCAE (*n* = 1), hypertension grade 3 CTCAE (*n* = 2), and concurrent proteinuria grade 3 CTCAE (*n* = 1)) on request of parents.

All adverse events recovered after appropriate treatment or after discontinuation of BVZ. After analyses of all blood samples obtained within the period of start of BVZ and 3 months after cessation, a decrease in neutrophil count grade 2 was found in 4 patients and grade 3 in 2 patients. All patients were concurrently treated with BVZ/IRI. No patient presented with any symptom related to a suspicion of infection. All patients received BVZ/IRI with no delay. Spontaneous recovery of neutrophil count occurred in all patients. These results are not included in the presentation of the toxicity profile.

### 3.5. Progression

Progression occurred in 21 patients (63.6%) after a median of 24.5 months (range 2–98 months) after start of BVZ. The increase of tumor volume in 18 patients (85.7%) and decrease of visual function in 3 patients (14,3%) were considered as progression by the local oncology team. The PFS curve is presented in Figure 1A (Kaplan–Meier plot). At 18 and 36 months after start of BVZ (6 and 24 months after intended cessation), PFS reduced from 70.9% (CI 54.8–87.0) to 38.0% (CI 20.3–55.7).

Progression occurred in eight of nine children with KIAA1549-BRAF fusion positive OPG; all were nNF1. In 7 of 13 NF1 patients, OPG progressed. The PFS curve of the 13 NF1 patients and 9 patients with KIAA1549-BRAF fusion are shown in Figure 1B. Comparison of PFS between both groups group showed a significant difference in PFS (log rank *p* < 0.01). No survival analysis has been performed on the nNF1 KIAA1549-BRAF fusion negative population, as the volume was too small (*n* = 3). Likewise in the nNF1 population with unknown KIAA1549-BRAF fusion status (*n* = 8), as fusion status was lacking.

The various treatment combinations (BVZ only/BVZ and IRI/BVZ and VBL) showed no difference among groups in PFS (log rank: *p* = 0.591). A trend towards a difference in PFS was observed among the diverse age categories (log rank: *p* = 0.083); however, group volumes were too small to reach significance.

Five out of twenty-one patients were on treatment when progression was observed, and sequential treatment was initiated. The median age at the start of this group was 1.2 years (range 0.9–8.2 year), and it contained 1 patient with NF1 (20%). One patient switched therapy prematurely as visual function did not improve and OPG tumor size did not reduce. Of the total population that progressed (*n* = 21), sequential therapy consisted of surgery (*n* = 2 (9.5%)), vinblastine (*n* = 2 (9.5%)), trametinib (*n* = 4 (19%), and BVZ restart (*n* = 12 (57.1%)) after an interval of median 8.5 months (range 4–75 months). The overall survival of the cohort was 97%: 1 infant (nNF1), diagnosed at 5 months old, died at the age of 2.7 years 6 months after the start of BVZ (third line SAT) due to progression.

## 4. Discussion

In this nationwide retrospective study on BVZ treatment for pediatric patients with OPG, the outcome was relatively effective as treatment resulted in the temporary stabilisation of tumor volume and the improvement of visual function in the majority of children. Radiological or clinical progression occurred in 21 patients (63.6%) after a median of 20 months after start of therapy. In contrast, visual function, represented by BCVA, remained stable in 74.4% of analysed eyes, and VF improved in 73.1%. Serious side effects occurred in almost one in six patients, but recovered after cessation of BVZ.

Multiple treatment episodes with SAT are a well-known necessity in subgroups of children with progressive OPG [10,28]. Since the publication of the first results on BVZ/IRI for LGG in 2009 [3], a total of 10 studies (in the English language) have reported on successive treatment with BVZ for progressive LGG in children. These studies included variable numbers of patients with LGG (range 7–88 patients), containing 2–88 OPG per cohort [3,6,7,8,9,10,11,12,13,14], and focused on radiologic response and toxicity profile. The current study exclusively evaluates progressive OPG treated with BVZ and mostly IRI, evaluating visual outcome in addition to PFS, radiologic tumor response, and toxicity.

In this study, OPG progression occurred during the intended 12 months of BVZ treatment in 15.1% of patients, but accumulated to 63.6% during follow-up after cessation of BVZ. The PFS at 18 and 36 months after start of treatment reduced from 70.9 to 38.0%, which is relatively similar to the study of Green at al. with a PFS of 29.0% at 36 months [10]. Previous reports on the rate of progression as an outcome parameter for treatment with BVZ for patients with LGG (in different cerebral locations) rendered a median of 45% (range 14–93% after 5–12 months of follow-up [6,7,8,9,11,12]. Our data support the moderate, but relatively temporary effectiveness of BVZ.

The relevance of tumor biology was evaluated via the KIAA1549-BRAF fusion status for progressive OPG, which showed a reduced PFS compared to NF1 OPG. Analyses on treatment of larger volumes of KIAA1549-BRAF fusion positive OPG would allow fusion subtype analyses [29], possibly revealing individual risk factors for the choice of optimal treatment. Nonetheless, as biopsy is often not performed due to the risk of further deterioration of visual functions, BVZ should currently be considered a feasible non-targeted alternative treatment.

Visual function can be seriously impaired in OPG. Therefore, testing is an essential component of the therapy evaluation of OPG. Visual acuity is currently considered as the main parameter of visual function evaluation in OPG.

Studies on BCVA, preferably monocular, as an outcome parameter for treatment evaluation require evaluation within a short time interval after cessation. Previous studies show that carboplatin-based first-order SAT results in monocular improvement of BCVA in 10–22% of patients, stabilization in 57–84%, and a decrease in 6–21% [17,30]. A study on progressive OPG treated with BVZ presented improvement in BCVA of 18.5%, stabilisation in 64.6%, and a decrease in 16.9% [10] In our study, BCVA stabilized or improved in 95% of analyzed eyes. Chiasmal and optic tract glioma mostly cause a combined decrease of BCVA and VF, but studies on VF as the outcome parameter for treatment of OPG are limited. Fisher et al. presented VF evaluation in 30 eyes after first order SAT, with stabilisation or improvement in 63% [31]. Fangusaro et al. reported on 19 patients treated with selumetinib, 26% of whom showed improvement and 74% of whom showed stabilisation of binocular VF [32]. Green et al. showed improvement of VF in 13%, stabilisation in 83%, and a decrease in 4% of OPG treated with BVZ [10]. In our study, monocular VF improved in 73.1% and stabilized in 15.4%. VF was obtained from analyses on 25 eyes of 16 patients only. VF testing was lacking in 17 patients due to blindness (*n* = 3) and lacking data (*n* = 14). Performing VF tests at a young age (<6 years) or in children with limited cooperation can be challenging with a high risk of bias. Limited cooperation due to young age, low vision, or cognitive impairment in NF1 children could account for this low frequency of VF reporting.

However, thus far, the relatively high level of improvement of VF justifies analysis in future trials. As BCVA and VF parameters did not fully change function similarly (improvement/stability/decrease) (Appendix A), combined analysis of BCVA and VF is warranted.

Previous studies on toxicity in BVZ treatment (CTCAE ≥ grade 3 for LGG) report an incidence of 0–29% during a treatment period of 10–24 doses of BVZ (10 m/kg every 2 weeks) [6,7,8,9,10,11,12,13]. In our study, side effects (15.2% of CTCAE ≥ grade 3) occurred early after the start of BVZ (and IRI/VBL) (median 9 weeks), which required cessation of BVZ (and IRI/VBL) in 83.4%. No side effect rendered long-term morbidity or mortality, as all functions recovered after appropriate treatment or cessation of BVZ and IRI. These results suggest an acceptable toxicity profile in relation to the clinical and radiologic stabilizing effect of BVZ. All registered side effects are not likely to be listed under adversities due to irinotecan or vinblastine, but they fit the BVZ toxicity profile. Therefore, the decreased neutrophil count (with no sign of infection), a known feature of treatment with IRI and not BVZ, was not included in the analyses. No data on long-term side effects were collected, as a majority of patients were treated by the shared care of pediatric specialists in different (non-academic) hospitals, which would possibly contribute to a bias in reports on side effects.

## 5. Strengths and Limitations

The strength of this study is the analysis of a nationwide group of children treated with BVZ for progressive OPG without selection bias. To minimize debate on diverse outcome parameters, we have clearly presented applied definitions of visual functions based on international consensus on published best available criteria in children with a brain tumor [23,27]. Although BCVA and VF were not obtained by a standard validated protocol, the data were carefully reviewed for outliers.

The limitation is found in the retrospective design of the study, which led to missing data on visual function, possible underreporting of side effects, irregularities in follow-up intervals, and diversity in applied MRI protocols.

At a young age, visual functions are part of continuing visual maturation. As currently change in VA function and radiologic response effects are not correlated [17], improvement of VA after treatment should be considered a combination of treatment effect and maturation of VA.

Furthermore, the learning curve for performing a VF with HFA, Goldmann, or Peritest is considerable for children. The high percentage of improvement of VF in this study could have a confounding effect on the individual learning process among patients, as VF test variability was not tested at the start of treatment with BVZ.

Radiologic response measurement in OPG encounters many challenges in volumetric evaluation. Although the RAPNO criteria on radiologic response assessment [15] are designed for pediatric LGG evaluation, manual three-directional measurements of diffuse OPG did not provide a reliable representation for response evaluation of this tumor with posterior offshoots and highly irregular contours. This inevitable problem necessitated solely chiasmal response evaluation in > 50% of diffuse OPG. The 25% volumetric change does not seem appropriate for OPG, as volumetric change ≥ 25% will be preceded by loss of visual function. Possibly, future semi-automated segmentation with extraction of tumor volume could assist in more detailed response assessment in OPG.

In conclusion, the treatment of BVZ for progressive OPG can be considered a relatively safe strategy, with temporary stabilization in a majority of patients. As visual function shows a high rate of improvement (in VF) and stabilisation (in BCVA), BVZ treatment is suggested to be a useful successive SAT. As the incidence of progressive OPG is low, its tissue analysis is scarce, and treatment with BVZ is rare, international retrospective and prospective collaboration is highly recommended to refine analysis of the treatment effect of BVZ in relation to NF1 status, anatomic location, and individual differences in tumor biology.

## Figures and Tables

**Figure 1 cancers-14-06087-f001:**
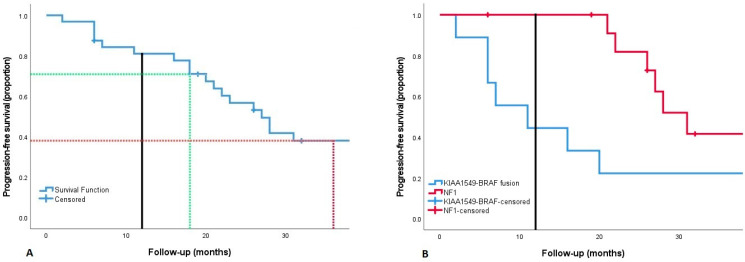
Kaplan–Meijer plot of progression-free survival for OPG treatment with BVZ. (**A**): Cumulative PFS overall population after start of treatment with BVZ for OPG: PFS reduces from 70.9% to 38.0% from 18 (green reference line) to 36 months (red reference line) after start (presumed 6 and 24 months after intended cessation of therapy). (**B**): Cumulative PFS between the NF1 (*n* = 13) and KIAA1549-BRAF fusion positive population (*n* = 9) (log rank: *p* < 0.01). Black line: reference line at 12 months: median completion of therapy cycles. Abbreviations: BVZ; bevacizumab, PFS: progression-free survival.

**Table 1 cancers-14-06087-t001:** Baseline characteristics of pediatric patients with OPG at start of treatment with BVZ.

Characteristics	Nr of Patients (%)
Total population	33
Sex	
Male	20 (60.6)
Female	13 (39.4)
NF1	13 (39.4)
Diagnosis based on clinical signs	6
DNA pathogenic mutation	7
**Anatomic location: stage MDC**	
MDC 2	4 (12.1)
MDC combined 1&2	4 (12.1)
MDC combined 2&3	154 (45.5)
MDC combined ≥ 3 stages	10 (30.3)
Hypothalamic involvement	28 (84.8)
Lepto-meningeal metastases	6 (18.2)
**Biopsy performed (NF1/nNF1)**	17 (3/14)
Obtained during surgery	9 (1/8)
Biopsy only	8 (2/6)
**Pathology**	17 (51.2)
Pilocytic astrocytoma	13
Fibrillary astrocytoma	2
Pilomyxoid astrocytoma	2
**Tumor biology (NF1/nNF1)**	13 (3/10)
BRAF V600E mutation (analysed in 12 samples)	0
KIAA1549-BRAF fusion (analysed in 15 samples)	9 (0/9) (8 of 9: PA)
**Age at diagnosis of OPG**	
Median (yr) (NF1/nNF1)	2.4 (5.2/1.1)
Range (yr)	0.3–10.2
NF1	2.4–10.2
nNF1	0.3–5.1
IQR (yr)	0.8–5.0
**Age at start of all therapy**	
Median (yr) (NF1/nNF1)	2.4 (5.6–1.2)
Range (yr)	0.3–16.0
NF1	2.4–16.0
nNF1	0.3–5.4
IQR (yr)	1.1–5.3
Interval diagnosis OPG-start 1th therapy	0.1 (0.0–9.3)
**Age at start of BVZ**	
Median (yr) (NF1/nNF1)	7.2 (11.0/4.2)
Range (yr)	0.7–17.7
NF1	4.7–17.7
nNF1	0.7–15.1
IQR (yr)	3.4–11.0
**Indication start BVZ**	
Radiologic progression	19 (57.6)
Radiologic progression and visual deterioration	1 (3.0)
Visual deterioration	12 (36.4)
New metastases	1 (3.0)
**BVZ initiation in SAT episode**	
1st	1 (3.0)
2nd	13 (39.4)
3rd	16 (48.5)
4th	3 (9.1)

Abbreviations: BRAF: the human gene that encodes the B-Raf protein; BVZ: bevacizumab; IQR: interquartile range; KIAA1549: human gene that encodes the KIAA1549 protein; MDC: Modified Dodge classification (stage 1: optic nerve(s); stage 2: chiasm; stage 3: optic tract; stage 4: posterior optic tract); NF1: Neurofibromatosis type 1; nNF1: no Neurofibromatosis type 1; PA: pilocytic astrocytoma; SAT: systemic anticancer therapy; yr: year.

**Table 2 cancers-14-06087-t002:** Baseline data on visual acuity and visual field and change in function after treatment with BVZ.

Visual Function & Change	
**BCVA at Start of BVZ (*n* = eyes)**	52
Bilateral blindness (*n* = patients)	3
Blind eyes	13
No data (*n* = per eye)	14
BCVA per eye ^1^ (*n* = eyes)	39
Median (LogMAR)	0.4
Range	−0.1–2.7
IQR	0.1–1.3
Binocular BCVA ^1^ at start BVZ (*n* = patients)	23
Median (LogMAR)	0.2
Range	−0.1–1.8
IQR	0.0–1.0
**BCVA after end BVZ (*n* = eyes)**	52
Bilateral blindness (*n* = patients)	3
Monocular blindness (*n* = eyes)	13
No data (*n*= eyes)	10
BCVA per eye ^1^ (*n* = eyes)	39
Median (LogMAR)	0.3
Range	−0.1–3.0
IQR	0.0–1.3
**Change in BCVA ^1^ (*n* = eyes)**	39
Median (LogMAR)	0
Range	−0.7–1.2
IQR	−0.1–0.02
Improvement (≤0.2 LogMAR)	8 (20.5%)
Stable (change within 0.2 LogMAR)	29 (74.4%)
Decrease (≥0.2 LogMAR)	2 (5.1%)
**Change in VF (*n* = eyes)**	26
Improvement	19 (73.1%)
Stable	4 (15.4%)
Decrease	2 (7.7%)
Shift ^2^	1 (3.8%)

BCVA is scored in LogMAR, Ad ^1^: After exclusion of blind eyes at start of BVZ, Ad ^2^: VF shift: VF loss in one quadrant, which on subsequent VF evaluation changes to VF loss in a different quadrant. Abbreviations: BCVA: Best Corrected Visual Acuity, IQR: interquartile range, VF: visual field.

**Table 3 cancers-14-06087-t003:** Side effects during treatment with bevacizumab.

Side Effect	Grade CTCAE	N (%)
Nausea	Grade 2	4 (12.1)
Hypertension	Grade 2	1 (3.0)
	Grade 3	3 (9.1)
Proteinuria	Grade 2	2 (6.1)
	Grade 3	1 (3.0)
Fatigue	Grade 2	2 (6.1)
Abdominal pain	Grade 3	1 (3.0)
Colitis	Grade 2	1 (3.0)
Gastric hemorrhage	Grade 3	1 (3.0)
Pneumonia	Grade 3	1 (3.0)

Side effects during treatment occurred in 12 patients: grade 2 and 3 according to CTCAE criteria (Common Terminology Criteria for Adverse Events) (version 5.0) [27]. All side effects recovered.

## Data Availability

The data presented in this study are available on request from the corresponding author. The data are not publicly available due to patients’ privacy protection.

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
