# Peer review of "Impact of Bevacizumab on Visual Function, Tumor Size, and Toxicity in Pediatric Progressive Optic Pathway Glioma: A Retrospective Nationwide Multicentre Study"

_cancers, 2022, doi:10.3390/cancers14246087_

Round 1
Reviewer 1 Report
The paper has a potential interest for readers, especially for the well done description of visual function.
I attach both the paper and the supplemental file with specific requirements and comments.
Since I cannot attach more than one file I copy here the table that I would like to be amended
Supplementary Table S2: Type of therapy applied per treatment phase for OPG.
|
Type of therapy per phase |
Nr of patients (%) |
|
SAT in episode 1 |
33 |
|
Carboplatin/vincristin |
30 (90.9) |
|
Carboplatin/vincristin/etoposide |
1 (3.0) |
|
Vinblastine |
1 (3.0) |
|
Bevacizumab/irinotecan |
1 (3.0) |
|
SAT in episode 2 |
32 |
|
Vinblastine |
18 (56.2) |
|
Vinblastine/carboplatin |
2 (6.2) |
|
Bevacizumab/irinotecan |
12 (37.5) |
|
SAT in phase 3 |
22 |
|
Temodal |
3 (13.6) |
|
Vinblastine |
2 (9.1) |
|
Bevacizumab/irinotecan |
17 (77.3) |
|
SAT in phase 4 |
1 |
|
Bevacizumab/irinotecan |
1 (3.0) |
|
Previous neurosurgical resection prior to BVZ |
9 |
|
1x resection |
8 (24.4) |
|
4x resection |
1 (3.0) |
|
Previous radiotherapy |
1 |
|
1x |
1 (3.0) |
Abbreviations: SAT: systemic antitumor therapy.

Author Response
Dear reviewer 1,
Please see the attachment.
Kind regards,
Carlien Bennebroek

Reviewer 2 Report
Dear authors,
your manuscript aim to evaluate the Impact of bevacizumab on visual function, tumor size, and toxicity in pediatric progressive optic pathway glioma. I think the manuscript can be of interest for journal audience however I have some comments:
Methods:
1. Did you include only incident users of bevacizumab? Please specify.
2. What is the date from which you start to following these patients? First bevacizumab use?
3. What are the censoring criteria? End of study period etc…
4. Do you have information also about overall survival?
5. Line 168: Did you search adverse events only in the period of treatment with bevacizumab?
Results
· Please provide some additional information about grade 4 hemorrhage.
· Please enhance quality of figure 1.
· I would avoid to report log rank test since you have very few patients in your cohort. If you would have a larger cohort I believe cox analysis would be more appropriate.
Discussion
As stated in a recently published meta-analysis of antiangiogenic drug in pediatric patients with solid tumors the evidence of safety profile of these drugs is limited only to acute toxicities (https://www.mdpi.com/2072-6694/14/21/5315). Do you have from your study any other information of long-term toxicities of bevacizumab? If no, please discuss it in the discussion section.
Author Response
Dear reviewer 2,
Please see the attachment.
Kind regards,
Carlien Bennebroek

Reviewer 3 Report
in brief this is a retrospective nationwide study which provides interesting and novel information mainly regarding visual fields and its possible correlation with visual acuity in this topic where similar papers with larger datasets (Green et Al, Neurooncology 2022) have recently published similar data I did not find any major flaw in this well written manuscript, and english language is ok. main suggestion would be to accept it after addition of latest publications (green et al) as well as the French Cohort published on J of Neuro-onc in the discussion section.
Author Response
Dear reviewer 3,
Please see the attachment.
Kind regards,
Carlien Bennebroek

Round 2
Reviewer 1 Report
There still a mistake with reference 13
RAPNO low-grade is exactly what I did alredy suggest (not high-grade as the authors tell in their response) i.e. Lancet Oncol 2020 Jun;21(6):e305-e316.
- doi: 10.1016/S1470-2045(20)30064-4
Reviewer 2 Report
Dear authors,
i have no further comments
